# The Obesogenic Gut Microbiota as a Crucial Factor Defining the Depletion of Predicted Enzyme Abundance for Vitamin B12 Synthesis in the Mouse Intestine

**DOI:** 10.3390/biomedicines12061280

**Published:** 2024-06-09

**Authors:** Anastasia A. Zabolotneva, Irina M. Kolesnikova, Ilya Yu. Vasiliev, Tatiana V. Grigoryeva, Sergei A. Roumiantsev, Aleksandr V. Shestopalov

**Affiliations:** 1Department of Biochemistry and Molecular Biology, Faculty of Medicine, N. I. Pirogov Russian National Research Medical University, 1 Ostrovitianov Str., Moscow 117997, Russia; sel_irka@list.ru (I.M.K.); rumyantsev.sergey@endocrincentr.ru (S.A.R.); al-shest@yandex.ru (A.V.S.); 2Laboratory of Biochemistry of Signaling Pathways, Endocrinology Research Center, 11 Dm. Ulyanova Str., Moscow 117036, Russia; mepk_m6@mail.ru (I.Y.V.); tatabio@inbox.ru (T.V.G.); 3Institute of Fundamental Medicine and Biology, Kazan Federal University, 18 Kremlyovskaya Str., Kazan 420008, Russia

**Keywords:** gut microbiota, obesity, cobalamin, vitamin b12, *db*/*db* mice

## Abstract

Currently, obesity is a critical global public health burden. Numerous studies have demonstrated the regulation of the pathogenesis of obesity and metabolic abnormalities by the gut microbiota and microbial factors; however, their involvement in the various degrees of obesity is not yet well understood. Previously, obesity has been shown to be associated with decreased levels of vitamin B12. Considering exclusive microbial production of vitamin B12, we hypothesized that a decrease in cobalamin levels in obese individuals may be at least partially caused by its depleted production in the intestinal tract by the commensal microbiota. In the present study, our aim was to estimate the abundance of enzymes and metabolic pathways for vitamin B12 synthesis in the gut microbiota of mouse models of alimentary and genetically determined obesity, to evaluate the contribution of the obesogenic microbiome to vitamin B12 synthesis in the gut. We have defined a significantly lower predicted abundance of enzymes and metabolic pathways for vitamin B12 biosynthesis in obese mice compared to non-obese mice, wherein enzyme depletion was more pronounced in *lepr*(−/−) (*db*/*db*) mice, which developed severe obesity. The predicted abundance of enzymes involved in cobalamin synthesis is strongly correlated with the representation of several microbes in high-fat diet-fed mice, while there were almost no correlations in *db*/*db* mice. Therefore, the degree of obesity and the composition of the correspondent microbiota are the main contributors to the representation of genes and pathways for cobalamin biosynthesis in the mouse gut.

## 1. Introduction

The last few decades have been marked by extensive research on the metabolic activity of the intestinal microbiota and its impact on host health [1]. The concept of an obesogenic microbiome, which is characterized by compositional and metabolic changes in the gut microbiota that are detrimental to the host, has been formed [2]. Although the characteristics of such an obesogenic microbiome may vary significantly, which is confirmed by contradictory results of microbiota analysis in different studies, some common trends have been described. Enrichment by enzymes for improved energy extraction from nutrients, decreased production of short chain fatty acids, and increased production of lipopolysaccharides stimulating proinflammatory response by the host immune system are some common signs of obesogenic microbiota [3]. Furthermore, it has been observed that the obese state of the host is associated with a decrease in the level of vitamin B12 in the blood, although the underlying mechanisms are unclear [4,5,6]. As shown in previous studies, vitamin B12 is a critical micronutrient associated with the foetal programming of obesity. Researchers from India and UK have revealed that low vitamin B12 status in the gestation period is associated with increased obesity and insulin resistance of the mother and birth weight and risk of insulin resistance and obesity in the offspring. In the study of obese adolescents, it was found that obesity has a 1.6-fold decreasing effect on vitamin B12 levels [7]. 

In preclinical studies, it was demonstrated that low levels of vitamin B12 could increase lipid accumulation in adipocytes and trigger dyslipidaemia in mice, suggesting that low levels of B12 and dyslipidaemia could be causally related [8]. A low serum B12 level has also been shown in people with nonalcoholic fatty liver disease, especially with grade 2 and 3 steatosis [9]. In the study of Boachie et al. [10], it was shown that low B12 in HepG2 cells increased the gene expression of lipogenesis and decreased lipid oxidation, resulting in increased intracellular triglycerides and the accumulation of lipid droplets.

It has been suggested that the primary cause of B12 malabsorption in certain human diseases is the gut microbiota’s depletion of vitamin B12 [11]. Cobalamin (vitamin B12) production is restricted to microorganisms, specifically anaerobes, and cannot be achieved by plants, animals, or fungi [12]. Although it is still not known whether the host uses microbially produced B12 in the gut [13], it was shown that human commensal gut microbes can produce extracellular vesicles containing multiple high-affinity vitamin B12 binding proteins, suggesting that the vesicles play a role in the elimination of micronutrients and their transfer to bacterial and host cells [14]. Many commensal gut bacteria have been indicated as vitamin B12 producers [15]. Vitamin B12 synthesis is a complex multistep process and requires a lot of enzymes (Figure 1).

We hypothesized that the decrease in cobalamin level in obese individuals may be at least partially caused by its depleted production by the commensal microbiota in the intestine. In our study, our aim was to estimate the abundance of enzymes and metabolic pathways for vitamin B12 synthesis in the gut microbiota of mouse models of alimentary and genetically determined obesity, to evaluate the contribution of the obesogenic microbiome to vitamin B12 synthesis in the gut. To this end, we performed a high-throughput metagenome sequencing analysis followed by a reconstruction of gut microbiota metabolic activity for high-fat diet-fed C57BL/6SPF mice (which are used as a model of alimentary obesity), standard diet-fed *db*/*db* mice (leptin receptor-deficient mice) (which are used as a model of genetically determined obesity), and standard diet-fed C57BL/6SPF mice (which are used as a control group).

## 2. Materials and Methods

### 2.1. Experimental Animals

C57BL/6SPF mice (n = 20, males) (bred at the Nursery of Laboratory Animals in Puschino, Puschino, Russia) and *db*/*db* (n = 10, males) mice (bred at JAX-East and JAX-West Nurseries of Laboratory Animals, Sacramento, CA, USA) were acclimated to housing conditions (22 °C, 55% humidity, 12 h:12 h light: dark cycle) in an SPF level animal center of I.M. Sechenov First Moscow State Medical University (Moscow, Russia) with ad libitum access to sterile food (Altromin 1324, Lage, Germany. The food includes 4.1% crude fat, 19.2% crude protein, 6.1% crude fiber, 5.9% crude ash, 53.4% nitrogen-free extractives, and 11.3% moisture) and water for 1 week prior to formal study. Following the adaption phase, the mice were split into groups based on genotype, with each group consisting of ten individual animals. The mice were 8 weeks old and weighed 19 ± 2 g on average at the start of the experiment. Laboratory animals were fed a high-fat diet enriched with triglycerides derived from animals, up to 30% of total calories (Altromin C 1090-30, Lage, Germany. The food includes 13.3% crude fat, 21.1% crude protein, 5.1% crude fiber, 3.9% crude ash, 50.8% nitrogen-free extractives, and 5.8% moisture), beginning at 8 weeks of age and continuing for 90 days until the end of the experiment. This allowed for the replication of the alimentary obesity model.

For the duration of the experiment, the mice in the *db*/*db* group and the animals in the control group (C57BL/6SPF) were fed a standard chow diet (Altromin 1324 FORTI, Lage, Germany). Mice were anesthetized with isofurane and euthanized after 90 days of feeding (RWD Life Science, Chenzhen, Guangdong, China). Sterile colon tissues were obtained. The tissue samples were immediately frozen in liquid nitrogen and stored at −80 °C until analysis. The colon samples were sent for high-throughput sequencing analysis after being divided into 1 cm sections under sterile conditions, deposited into different sterile Eppendorf tubes, and maintained on dry ice (10 samples for each group) (Figure 2).

All animal experiments were approved by the Ethics Committee for Animal Research, I.M. Sechenov First Moscow State Medical University, Russia (protocol number 96 from 2 September 2021).

### 2.2. Measurement of Serum Insulin, Leptin, and Adiponectin Levels

The mice fasted for 8 h. Blood samples were collected after fasting from anaesthetized mice using the eyeball enucleation method. Blood was collected in sterile collection tubes immediately. Serum was spun down at 8000× *g* for 8 min at 4 °C to remove remaining cellular debris. Insulin concentration was measured using the Mouse Insulin ELISA Kit (Abcam, ab277390), leptin concentration was measured using Duo Set ELISA Development system Mouse Leptin (R&DSystem, DY495-05), and adiponectin concentration was measured using Adiponectin/Acrp30 DuoSet ELISA (R&DSystem, DY1119) according to the manufacturer’s instructions.

### 2.3. Measurement of Fasting Blood Glucose Level and HOMA-IR

Fasting levels of glucose were measured every 7 days for every group of mice; mouse tail snip blood samples were used for the analysis. Blood glucose was measured using an ACCU-CHEK Aviva glucometer (Roche, Switzerland). The animals fasted for 5 h with free access to water before a fasting blood glucose test. The HOMA-IR (homeostasis model assessment of insulin resistance) index was calculated as [fasting serum glucose × fasting serum insulin/22.5] to assess insulin resistance.

### 2.4. Histopathology of Mouse Liver, Pancreas, Skeletal Muscle, and Adipose Tissues

Organ fixation after collection was carried out for at least 72 h. After fixation, the organs were dehydrated, soaked in paraffin, and cut into 4–5-micron-thick sections. Sections were stained with hematoxylin-eosin and examined by light microscopy (magnification ×10, ×40). The research was carried out using Leica (Germany) histological equipment.

### 2.5. High-Throughput Sequencing Analysis and Reconstruction of Intestinal Microbiota Metabolic Activity

The scientific research laboratory «Multiomics technologies of living systems» (Kazan, Russia) performed the microbiota analysis. Mouse stool samples were treated with the FastDNATM Spin Kit for Faeces (MP Biomedicals, Santa Ana, CA, USA) to extract genomic DNA. The bacterial 16S rRNA gene’s V3–V4 region was amplified using specific primers (see Appendix A). The PCR product purification process using AMPure XP Beads (Beckman Coulter, Brea, CA, USA, CB55766755) to barcode each sample was followed by secondary-round PCR amplification utilizing index primers. Using a Qubit 2.0 Fluorometer and the Qubit dsDNA High Sensitivity Assay Kit (Invitrogen, Carlsbad, CA, USA), the concentration of amplicons was determined. Prior to sequencing, the samples were mixed in an identical mole ratio to complete the sample preparation.

The libraries were then high-throughput sequenced (2 × 300 bp reads) (Illumina Miseq, Illumina, CA, USA). The raw reads were processed using QIIME2 v2023.7.0 [16] and PICRUSt2 v2.5.2 software [17] (accessed on 12 September 2023). According to the results of data sequencing using PICRUSt2 v2.5.2 software, the microbial metabolic pathways encoded by the detected bacterial genomes were scored, and the most abundant pathways were detected by multiple *t*-test analysis.

### 2.6. Statistical Data Analysis

Statistical processing of the data was carried out using the method of nonparametric statistics using the GraphPad Prism 10 v10.0.2 (171) statistical software package. The mean and standard deviation were used to present all data. All in vivo experimental data were analyzed using Welch’s one-way analysis of variance (ANOVA) or multiple Mann–Whitney tests using the two-stage step-up method (Benjamini, Krieger, and Yekutieli) (false discovery rate Q = 5%). *p*-values less than 0.05 were considered statistically significant (* *p* < 0.05, ** *p* < 0.01, *** *p* < 0.001). A correlation analysis was performed according to Spearman with an assessment of the statistical significance of the correlation coefficient.

## 3. Results

### 3.1. Body Weight and Hormonal Status

Three groups of mice (males, n = 10 for each group) were fed a standard or high-fat diet for 90 days. The dynamics of weight gain is shown in Figure 3a. At the end of the experiment, the final weights of the mice were compared (Figure 3b). Mice in both obesity models showed significantly higher body weight compared to the control group. The mean weight of the mice in control group was 28.5 ± 2.5 g; in the high-fat diet-fed mice, the mean weight was 34.7 ± 1.4 g (weight gain is 17.9% compared to the control group); and in *db*/*db* mice, the mean weight was 59.2 ± 4.1 g (weight gain is 51.9% compared to the control group).

Although there are no strict criteria for the development of alimentary obesity in mice, it is generally accepted that 20–30% weight gain points to reaching the obesity state [18]. In our experiment, weight gain in a high-fat diet mouse model group was 17.9% higher than in the control group, which was not enough to develop complications of severe obesity. However, such weight gain corresponds to stage 1 of obesity and, most importantly, these mice demonstrated dysmetabolic characteristics associated with obesity, as shown below. On the contrary, *db/db* mice gained 51.9% more weight compared to the control group, which indicates the development of severe obesity.

To confirm the genetically determined character of obesity, we compared insulin, leptin, and adiponectin levels in the blood serum of mice in all experimental groups. We established that insulin (Figure 4a) and leptin (Figure 4b) levels were significantly higher in *db*/*db* mice compared to C57Bl6 mice (15-times higher for insulin and 21-times higher for leptin), while the level of adiponectin decreased in mice fed a high-fat diet but not in *db*/*db* mice (Figure 4c). Furthermore, the level of leptin in high-fat diet-fed mice also increased compared to the control group, but this was not as crucial as in *db*/*db* mice. Significantly increased levels of insulin and leptin in *db*/*db* mice confirmed strong insulin resistance, which was developed due to leptin receptor deficiency. The increased concentration of leptin and decreased concentration of adiponectin in high-fat diet-fed mice also evidences the dysmetabolic state of the animals.

To investigate insulin resistance in mice, we conducted glucose dynamic observation and evaluated HOMA-IR indexes in all experimental groups. Glucose levels and HOMA-IR were significantly higher in *db/db* mice compared to the control group but did not change in HFD-fed mice (Figure 5a,b).

We also investigated the morphological state of the livers of all experimental mice. All mice in the HFD-fed group (Figure 6e–h) and *db*/*db* group developed liver steatosis (fatty liver) (Figure 6i–l), in contrast to SD-fed C57Bl6 mice, which demonstrated normal liver morphology without fat and inflammatory infiltration (Figure 6a–d). As expected, fat infiltration of the liver in *db*/*db* mice was significantly more profound than in HFD-fed mice.

Therefore, significant differences in body weight, leptin and adiponectin levels and hepatosteatosis developed in the HFD-fed and *db/db* groups of mice can approve the presence of metabolic changes associated with diet or genetic background, although insulin resistance, as expected, was developed in the group of *db*/*db* mice only. Thus, we can consider the HFD-fed group of mice as an overweight group with slight metabolic complications, while *db/db* mice simulated severe obesity with strong metabolic disturbances.

### 3.2. Microbiota Analysis

#### 3.2.1. Taxonomy Composition and Alpha Diversity

To characterize the composition of the mouse gut microbiota, we performed a high-throughput metagenome sequencing analysis of 30 microbiota samples taken from all experimental animals. After OTU annotation, we compared the differences in alpha diversity and microbial community structures. We established that the alpha diversity indexes (observed OTUs, PD whole tree, Chao1, Simpson, and Shannon) were significantly lower (*p* < 0.05) in *db*/*db* mice compared to the control group (Figure 7a–e), indicating a decrease in the richness and diversity of microbial species in severely obese mice. The microbiota of mice fed a high-fat diet also showed a decrease in alpha diversity; however, only PD whole-tree and Shannon indexes were lower in the BL/HFD group compared to the control group (Figure 7b,e).

We also investigated the taxonomic structure of bacterial communities (Figure 8a and Figure 9a–c). At the phylum level, we observed a significant decrease in the representation of Verrucomicrobia and Actinobacteria in *db*/*db* mice compared with HFD- and SD-fed mice (Figure 8b,c). Furthermore, the representation of Bacillota in *db*/*db* mice was higher than in the HFD feeding group (Figure 8b).

At the family level, we observed a significant increase in the representation of Prevotellaceae with a simultaneous decrease in the representation of Verrucomicrobiaceae in the microbiota of *db*/*db* mice compared to HFD-fed mice (Figure 9a–c). These changes are known to be associated with an obese phenotype [19], although the representation of Verrucomicrobiaceae increased in HFD-fed mice compared to SD-fed mice.

At the species level, we found an increase in *Bacteroides acidifaciens* in *db*/*db* mice compared to BL/SD and BL/HFD mice, while the representation of bacteria beneficial for host health (*Bifidobacterium pseudolongum*, *Akkermansia muciniphila*, *Adlercreutzia sp.*) decreased in *db*/*db* mice compared only to HFD-fed mice, not SD-fed mice (Table 1).

All these findings confirm the existence of specific obesogenic changes in the gut microbiota communities of mice, which is more pronounced in *db*/*db* mice than in HFD-fed mice, which is in concordance with the higher body weight and hormonal disturbances of *db*/*db* mice.

#### 3.2.2. Reconstruction of the Metabolic Activity of the Mouse Gut Microbiota and Representation of the Pathways Responsible for Vitamin B12 Biosynthesis

To establish whether enzymes and metabolic pathways for B12 synthesis are less abundant in obese mice compared to mice fed a standard diet, we carried out a reconstruction of microbiota metabolic activity by using a PICRUSt2 analytic tool based on metagenome sequencing data analysis, which allowed us to estimate the predicted abundance of bacterial genes in mouse gut microbial communities.

According to the results of PICRUSt2, among the 423 metabolic pathways analyzed, we revealed some differences in the abundance of the cobalamin synthesis pathways of mice fed *db*/*db* and an HFD compared to C57Bl6 mice fed an SD. Namely, the representation of six pathways for vitamin B12 synthesis in *db*/*db* mice and four pathways in C57Bl6 mice fed an HFD was decreased (Table 2).

#### 3.2.3. Abundance of Enzymes for Vitamin B12 Synthesis in Gut Bacteria According to the Results of the Metagenome Sequencing

PICRUSt2 analysis was also used to estimate the abundance of enzymes involved in cobalamin synthesis. To this aim, we chose 37 enzymes required for cobalamine synthesis (Figure 1) among more than 8000 enzymes represented in investigated microbiomes. Only three enzymes (EC 1.16.1.4, EC 1.16.1.6, [EC 2.1.2.272]) were not presented in our data and were not investigated.

We found that the representation of almost all enzymes (except for adenosylcobalamine hydrolase and vitamin B12-transporting ATPase) needed for vitamin B12 synthesis was decreased in *db*/*db* mice compared to C57Bl6 mice fed an SD (Table 3), while there was no difference in enzyme representation in other comparisons.

These observations may point to depletion of vitamin B12 synthesis in the gut of obese mice, mainly in *db*/*db* mice.

### 3.3. Correlation Analysis

To establish the role of individual species in cobalamin synthesis, we conducted Spearman correlation analysis, which revealed some important trends. We noticed that the abundance of enzymes involved in cobalamin biosynthesis is strongly correlated in a positive and negative way with the representation of several species in the microbiota of C57Bl6 mice fed a high-fat diet (Table 4). For example, we observed strong negative correlations between the representation of *Akkermansia sp.* and strong positive correlations between the representation of Lachnospiraceae bacteria and most of the enzymes issued. According to our results, bacteria of Verrucomicrobia phyla were represented significantly more in the microbiota of HFD-fed mice but not in SD-fed mice. Furthermore, it has previously been shown that in obese people, dietary vitamin B12 intake was inversely correlated with *Akkermansia muciniphila* species and species of the Verrucomicrobia phylum, while it was positively correlated with Bacteroides species [20].

At the same time, we observed almost a complete loss of correlations in the microbiota of *db*/*db* mice, except for positive correlations for species of the Lachnospiraceae family (Table 5). This fact is apparently caused by a significant decrease in the alpha diversity of microbiota communities and a loss of microbes involved in cobalamine synthesis.

## 4. Discussion

Vitamin B12 is an essential component for pro- and eucaryotic living organisms [21]. Because it is synthesized exclusively in bacterial cells, it can be accumulated in animals that receive it with food [22]. Many species of commensal bacteria inhabiting the mammalian gut are capable of synthesizing vitamin B12; however, it is not reliably known whether this microbiota-derived vitamin can be effectively absorbed into the host’s circulation and contribute to the systemic level of cobalamin [23]. In mammalian cells, vitamin B12 is used for the methionine synthase reaction and for the metabolism of amino and fatty acids [24]. Vitamin B12 deficiency is associated with the development of megaloblastic anemia and neurological complications [23]. Furthermore, an obese state and other dysmetabolic conditions have been found to be associated with decreased levels of vitamin B12 [6]. The fact that obesity is coupled with disturbances in intestinal microbiota composition, and metabolic activity also points to possible disturbances in cobalamin synthesis by commensal microbes in obesity. Interestingly, there may be a reciprocal relationship between the gut microbiota and vitamin B12 levels [15]. Vitamin B12 deficiency can alter the balance and functional interaction of the gut microbial community [25]. Therefore, by changing the microbial composition of the gut, interactions between vitamin B12 and the gut microbiota can prevent the development of obesity [26]. Thus, there may be a relationship between obesity, intestinal flora, and vitamin B12 levels [24].

To our knowledge, there are few studies investigating the ability of the intestinal microbiota to synthesize vitamin B12 in association with obesity.

Mice deficient in the leptin receptor (*db*/*db*) develop severe obesity with corresponding hormonal changes, such as hyperinsulinemia and hyperleptinemia, and they are widely used as a model of hyperphagia, obesity, and type 2 diabetes mellitus [27]. Furthermore, as we have shown in the present investigation, the gut microbiota of *db*/*db* mice differ significantly compared to the microbiota of C57Bl6 mice. That is, we observed a dramatic decrease in alpha diversity and the representation of ‘obesoprotective’ microbes, such as *Akkermansia miciniphila* and *Bifidobacterium pseudolongum,* along with an increase in the abundance of ‘obesogenic’ *Prevotella* in *db*/*db* mice, in contrast to C57Bl6 mice fed a high-fat diet. Mice that received an HFD showed an increased representation of Verrucomicrobia and Actinobacteria phyla, which may be associated with increased fuel sources for bacterial growth. However, the alpha diversity of the gut microbiota community of HFD-fed mice decreased as well, as is confirmed by the Shannon and PD whole-tree indexes.

For the first time, we have shown that severe obesity with corresponding metabolic disturbances, which are developed in *db*/*db* mice, is associated with a decrease in the representation of enzymes and metabolic pathways for the synthesis of cobalamin. However, C57Bl6 mice were fed a high-fat diet and also developed obesity, and although they gained less weight than *db*/*db* mice, they showed no differences in enzyme abundance and fewer pathways of vitamin B12 synthesis. Furthermore, strong associations between the representation of gut microbes and an abundance of cobalamin biosynthetic enzymes were described for HFD-fed mice. For example, we have seen strong negative correlations for the genera Colidextribacter, Akkermansia, Bacteroides, and Faecalibacterium, while the genera Muribaculaceae, Enterococcus, Butyricoccus and the uncultured genus of the Lachnospiraceae family were negatively associated with enzyme abundance. On the other hand, in *db*/*bd* mice, only positive correlations were observed with the uncultured genus of the Lachnospiraceae family, whereas other microbes showed no or unitary correlations with enzyme abundance.

Therefore, the degree of obesity and the composition of the correspondent microbiota are the main contributors to the predicted abundance of genes and pathways for cobalamin biosynthesis in the gut of mice. It can be proposed that the ‘obesogenic’ microbiota community is dwindling with vitamin B12 microbe producers, which can lead to the formation of a more aberrant microbial community and lower levels of vitamin B12 in the host. However, this hypothesis should be confirmed by studying serum levels of cobalamin in association with the degree of obesity and intestinal microbiota composition, as well as the direct contribution of separate microbes in cobalamin synthesis.

## 5. Conclusions

In this study, we performed a high-throughput metagenome sequencing analysis followed by a reconstruction of the metabolic activity of the gut microbiota for a high-fat diet-fed C57BL/6SPF mice, standard diet-fed *db*/*db* mice, and standard diet-fed C57BL/6SPF mice. We observed a specific obesogenic shift in mouse gut microbiota communities, which was more pronounced in *db*/*db* mice than in HFD-fed mice and was consistent with higher body weight and hormonal disturbances of *db*/*db* mice. We have defined a significantly lower predicted abundance of enzymes and metabolic pathways for vitamin B12 biosynthesis in obese mice compared to non-obese mice, where enzyme depletion was more pronounced in *db*/*db* mice, which developed severe obesity. The abundance of enzymes involved in cobalamin synthesis is strongly correlated with the representation of several microbes in HFD-fed mice, whereas there were almost no correlations in *db*/*db* mice. Therefore, the obesogenic gut microbiota can be implicated in decreased vitamin B12 synthesis in the gut, which, in turn, can influence host levels of cobalamin and make obese mice more prone to the development of cobalamin deficiency. However, because these findings are based on bioinformatic analysis, more direct molecular experiments are needed that confirm the involvement of the gut microbiota in vitamin B12 synthesis in connection with host vitamin B12 status and obesity development.

## Figures and Tables

**Figure 1 biomedicines-12-01280-f001:**
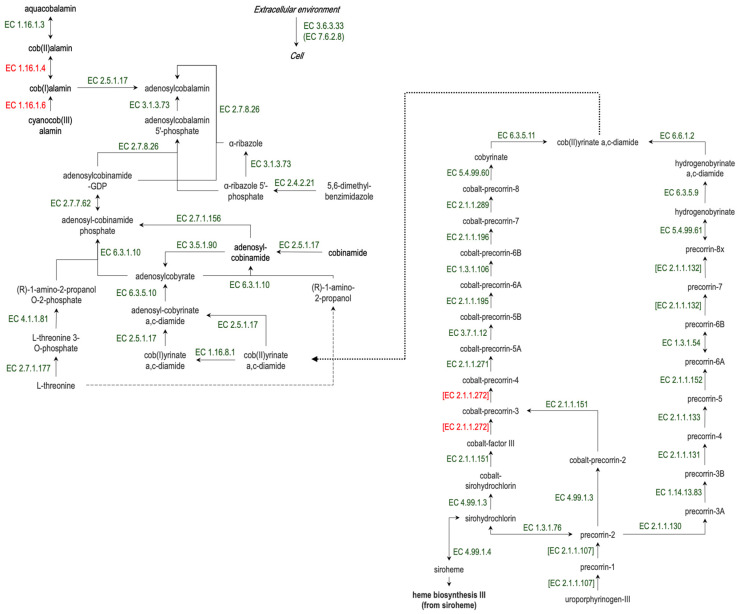
The scheme of vitamin B12 synthesis in bacteria. The predicted abundance of all enzymes, except EC 1.16.1.4, 1.16.1.6, [2.1.1.272] (red font), was investigated in the present study.

**Figure 2 biomedicines-12-01280-f002:**
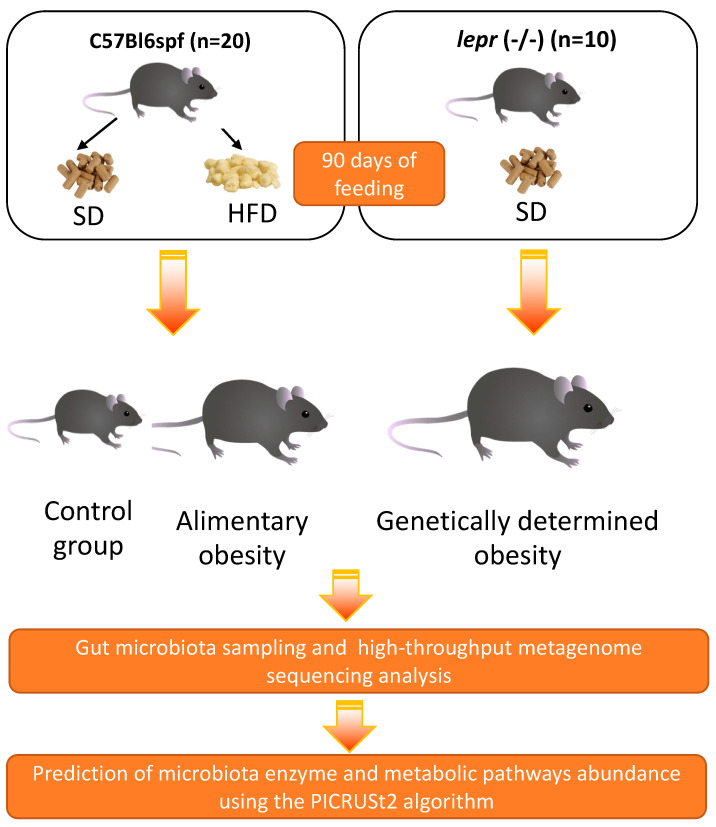
Experimental design showing the distribution of mice into study groups. SD, standard diet; HFD, high-fat diet.

**Figure 3 biomedicines-12-01280-f003:**
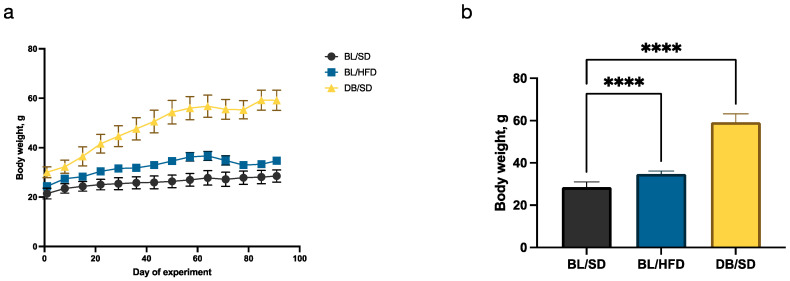
Changes in mouse body weight during the course of the experiment: (**a**)—mouse BW dynamics during the 90-day feeding experiment in different study groups; (**b**)—ANOVA of mouse BW in different experimental groups did not reveal any differences after 90 days of feeding (**** *p* < 0.0001). BL/SD, C57Bl6/spf mice fed a standard diet; BL/HFD, C57Bl6/spf mice fed a high-fat diet; DB/SD, *db*/*db* mice fed a standard diet.

**Figure 4 biomedicines-12-01280-f004:**
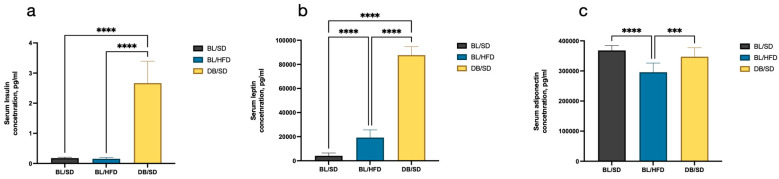
Results of serum insulin (**a**), leptin (**b**) and adiponectin (**c**) levels in different groups of mice at the end of the experiment. Comparisons were carried out using one-way analysis of variance followed by Tukey’s multiple comparison test (*** *p* < 0.001, **** *p* < 0.0001). BL/SD, C57Bl6/spf mice fed a standard diet; BL/HFD, C57Bl6/spf mice fed a high-fat diet; DB/SD, *db*/*db* mice fed a standard diet.

**Figure 5 biomedicines-12-01280-f005:**
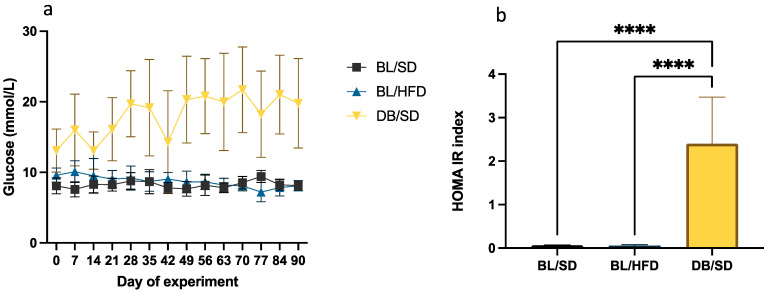
Serum glucose dynamics (**a**) and comparative analysis of HOMA-IR indexes (**b**) observed in different groups of experimental mice (**** *p* < 0.0001). BL/SD, C57Bl6/spf mice fed a standard diet; BL/HFD, C57Bl6/spf mice fed a high-fat diet; DB/SD, *db*/*db* mice fed a standard diet.

**Figure 6 biomedicines-12-01280-f006:**
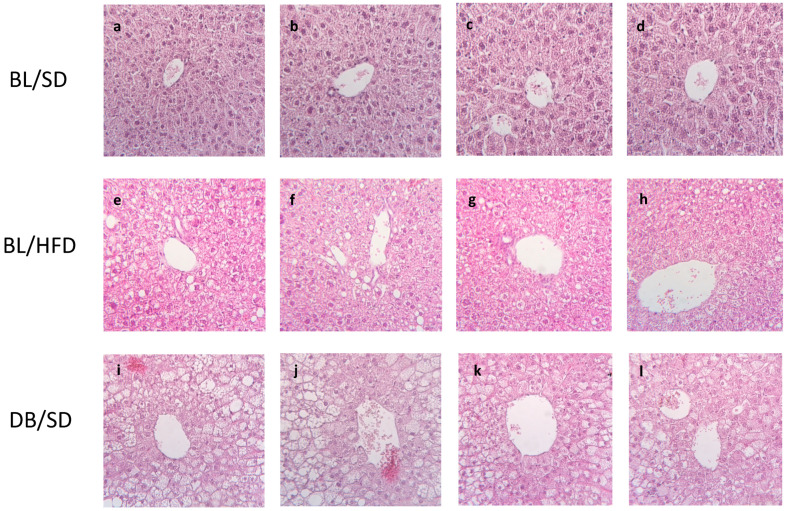
Liver sections of the control group of mice (**a**–**d**), C57Bl6/spf mice fed a high-fat diet (**e**–**h**), *db*/*db* mice fed a standard diet (**i**–**l**). The portal vein surrounded by hepatocytes in liver lobe is presented in every microphotograph. Fat infiltration is detected as empty vacuoles situated between hepatocytes. BL/SD, C57Bl6/spf mice fed a standard diet; BL/HFD, C57Bl6/spf mice fed a high-fat diet; DB/SD, *db*/*db* mice fed a standard diet.

**Figure 7 biomedicines-12-01280-f007:**
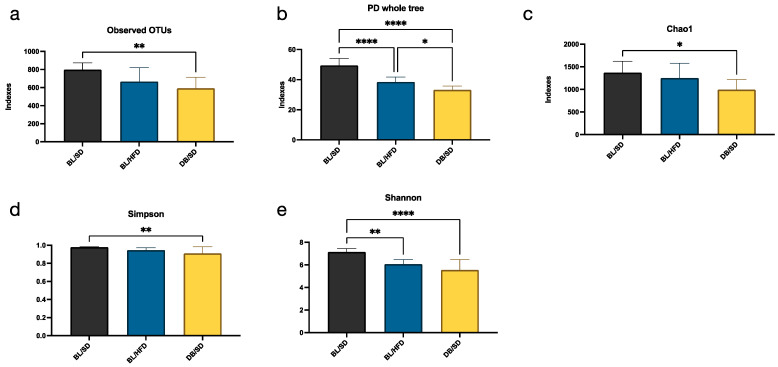
Results of one-way ANOVA followed by Tukey’s multiple comparison test for alpha diversity indices ((**a**)—for observed OTUs, (**b**)—for PD whole tree, (**c**)—for Chao1, (**d**)—for Simpson, (**e**)—for Shannon), in different groups of mice. * *p* < 0.05, ** *p* < 0.01, **** *p* < 0.0001. BL/SD, C57Bl6/spf mice fed a standard diet; BL/HFD, C57Bl6/spf mice fed a high-fat diet; DB/SD, *db*/*db* mice fed a standard diet.

**Figure 8 biomedicines-12-01280-f008:**
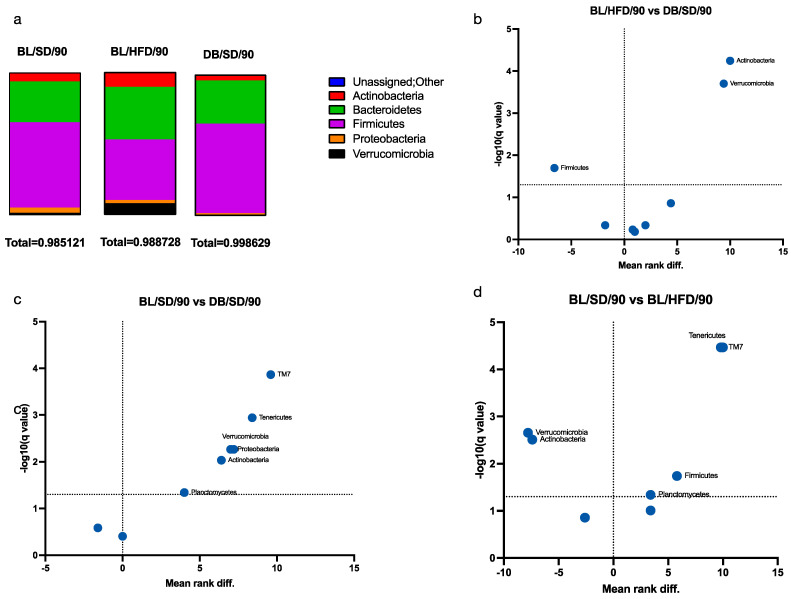
Taxonomy structure of mouse gut microbial communities at the phylum level: (**a**)—diagrams of comparative phyla representation in three groups of mice; (**b**)—volcano plot demonstrating differences in the representation of bacterial phyla in C57Bl6 mice fed a standard or a high-fat diet; (**c**)—volcano plot demonstrating differences in the representation of bacterial phyla in C57Bl6 and *db/db* mice fed a standard diet; (**d**)—volcano plot demonstrating differences in the representation of bacterial phyla in C57Bl6 mice fed a high-fat diet and in *db*/*db* mice fed a standard diet. The Q value reflects a false discovery rate of 5%. The mean rank difference values reflect the direction of changes in the abundance of bacterial phyla (values less than zero indicate an increased representation of phyla, while values greater than zero indicate a decreased representation of phyla in the microbiota of *db*/*db* mice (**b**,**c**) or HFD-fed mice (**d**). Statistically significant values (*p* < 0.01) are indicated. BL/SD, C57Bl6/spf mice fed a standard diet; BL/HFD, C57Bl6/spf mice fed a high-fat diet; DB/SD, *db*/*db* mice fed a standard diet.

**Figure 9 biomedicines-12-01280-f009:**
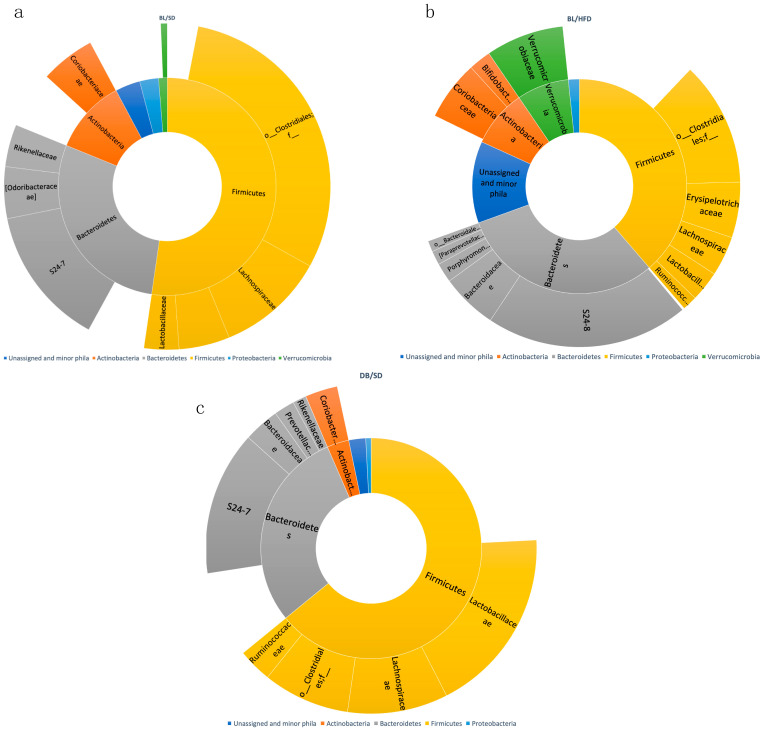
Taxonomy structure of mouse gut microbial communities at family level: (**a**)—diagram of families’ representation in C57Bl6 mice fed a standard diet; (**b**)—diagram of families’ representation in C57Bl6 mice fed a high-fat diet; (**c**)—diagram of families’ representation in *db*/*db* mice fed a standard diet. BL/SD, C57Bl6/spf mice fed a standard diet; BL/HFD, C57Bl6/spf mice fed a high-fat diet; DB/SD, *db*/*db* mice fed a standard diet.

**Table 1 biomedicines-12-01280-t001:** Differentially represented bacterial species in different groups of mice according to multiple Mann–Whitney tests with the two-stage step-up method (Benjamini, Krieger, and Yekutieli) (false discovery rate Q = 5%). *p* values less than 0.05 were considered to indicate statistical significance. a. Comparison in BL/HFD and DB/SD groups; arrows show increased or decreased microbe representation in *db*/*db* mice; b. Comparison in BL/SD and DB/SD groups; arrows show increased or decreased microbe representation in *db*/*db* mice; c. Comparison in BL/SD and BL/HFD groups; arrows show increased or decreased microbe representation in HFD-fed mice. BL/SD, C57Bl6/spf mice fed a standard diet; BL/HFD, C57Bl6/spf mice fed a high-fat diet; DB/SD, *db*/*db* mice fed a standard diet.

a. BL/HFD vs. DB/SD	b. BL/SD vs. DB/SD	c. BL/SD vs. BL/HFD
*Bifidobacterium pseudolongum*				*Bifidobacterium pseudolongum*	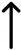
*Bacteroides sp.*				*Bacteroides sp.*	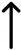
*Parabacteroides distasonis*		*Parabacteroides distasonis*		*Parabacteroides distasonis*	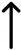
*Allobaculum sp.*				*Allobaculum sp.*	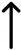
*Akkermansia muciniphila*				*Akkermansia muciniphila*	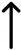
*Streptococcus sp.*		*Streptococcus sp.*			
*Adlercreutzia sp.*					
*Clostridium sp.*		*Clostridium sp.*		*Clostridium sp.*	
*Ureaplasma sp.*					
*Anaeroplasma sp.*	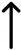			*Anaeroplasma sp.*	
*Bacteroides acidifaciens*	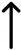	*Bacteroides acidifaciens*	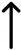		
*Anaerotruncus sp.*	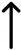				
*Ruminococcus sp.*	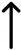			*Ruminococcus sp.*	
*Prevotella sp.*	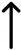				
*Rikenella sp.*	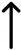			*Rikenella sp.*	
		*AF12 sp.*		*AF12 sp.*	
		*Coprobacillus sp.*		*Coprobacillus sp.*	
		*Dorea sp.*		*Dorea sp.*	
		*Sutterella sp.*		*Sutterella sp.*	
		*Odoribacter sp.*		*Odoribacter sp.*	
		*[Ruminococcus] gnavus*			
				*Enterococcus sp.*	
				*Coprococcus sp.*	
				*Parabacteroides sp.*	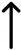

**Table 2 biomedicines-12-01280-t002:** Differentially represented metabolic pathways involved in vitamin B12 synthesis according to the results of multiple Mann–Whitney tests with the two-stage step-up method (Benjamini, Krieger, and Yekutieli) (false discovery rate Q = 5%). *p-*values less than 0.05 were considered to indicate statistical significance.

Pathways with Decreased Representation in *db*/*db* Mice Fed a Standard Chow Diet Compared to a Control Group (*p* < 0.01, q < 0.01)	Pathways with Decreased Representation in C57Bl6/spf Mice Fed a High-Fat Diet Compared to a Control Group (*p* < 0.001, q < 0.001)
cob(II)yrinate a,c-diamide biosynthesis I (early cobalt insertion) adenosylcobalamin biosynthesis I (early cobalt insertion) adenosylcobalamin biosynthesis II (late cobalt incorporation) adenosylcobalamin salvage from cobinamide I adenosylcobalamin biosynthesis from cobyrinate a,c-diamide I adenosylcobalamin salvage from cobinamide II	adenosylcobalamin biosynthesis II (late cobalt incorporation) adenosylcobalamin salvage from cobinamide I adenosylcobalamin biosynthesis from cobyrinate a,c-diamide I adenosylcobalamin salvage from cobinamide II

**Table 3 biomedicines-12-01280-t003:** Differentially represented enzymes involved in vitamin B12 synthesis in different groups of mice according to the results of multiple Mann–Whitney tests with the two-stage step-up method (Benjamini, Krieger and Yekutieli) (false discovery rate Q = 5%). *p* values less than 0.05 were considered to indicate statistical significance. a. Comparison in BL/SD and BL/HFD groups; red cells indicate decreased enzyme representation in HFD-fed mice; b. Comparison in BL/SD and DB/SD groups; red cells indicate decreased enzyme representation in *db*/*db* mice; c. Comparison in BL/HFD and DB/SD groups; red cells indicate decreased enzyme representation in *db*/*db* mice. BL/SD, C57Bl6/spf mice fed a standard diet; BL/HFD, C57Bl6/spf mice fed a high-fat diet; DB/SD, *db*/*db* mice fed a standard diet.

Enzymes	a. SD vs. HFD	b. SD vs. DB	c. HFD vs. DB
Precorrin-3B synthase			
Aquacobalamin reductase			
Cob(II)yrinic acid a,c-diamide reductase			
Cobalt-precorrin-6A reductase			
Precorrin-6A reductase			
Precorr in-2 dehydrogenase			
Uroporphyrinogen-III C-methyltransferase			
Precorrin-2 C(20)-methyltransferase			
Precorrin-3B C(17)-methyltransferase			
Precorrin-6B C(5,15)-methyltransferase (decarboxylating)			
Precorrin-4 C(11)-methyltransferase			
Cobalt-factor II C(20)-methyltransferase			
Precorrin-6A synthase (deacetylating)			
Cobalt-precorrin-5B (C(1))-methyltransferase			
Cobalt-precorrin-6B (C(15))-methyltransferase (decarboxylating)			
Cobalt-precorrin-4 methyltransferase			
Cobalt-precorrin-7 (C(5))-methyltransferase			
Nicotinate-nucleotide--dimethylbenzimidazole phosphoribosyltransferase			
Cob(I)yrinic acid a,c-diamide adenosyltransferase			
Adenosylcobinamide kinase			
L-threonine kinase			
Adenosylcobinamide-phosphate guanylyltransferase			
Adenosylcobinamide-GDP ribazoletransferase			
Adenosylcobalamin/alpha-ribazole phosphatase			
Adenosylcobinamide hydrolase			
Vitamin B12-transporting ATPase			
Cobalt-precorrin 5A hydrolase			
Threonine-phosphate decarboxylase			
Sirohydrochlorin cobaltochelatase			
Sirohydrochlorin ferrochelatase			
Cobalt-precorrin-8 methylmutase			
Precorrin-8X methylmutase			
Adenosylcobinamide-phosphate synthase			
Adenosylcobyric acid synthase (glutamine-hydrolyzing)			
Cobyrinate a,c-diamide synthase (glutamine-hydrolyzing)			
Hydrogenobyrinic acid a,c-diamide synthase (glutamine-hydrolyzing)			
Cobaltochelatase			

**Table 4 biomedicines-12-01280-t004:** Spearman correlations evaluated for the representation of bacteria species in microbiota of HFD-fed mice and enzymes for cobalamin biosynthesis according to the data of metagenome sequencing. Green font shows positive correlations (*p* < 0.05), red font shows negative correlations (*p* < 0.05), black font shows non-significant correlations (*p* > 0.05). All bacteria species were uncultured bacteria, except of Enterococcus sp. and Lactobacillus plantarum.

	g__Bacteroides	g__Muribaculaceae	g__Muribaculaceae	g__Faecalibaculum	s__Enterococcus_sp.	s__Lactobacillus_plantarum	f__Lachnospiraceae	f__Lachnospiraceae	g__Butyricicoccus	g__Colidextribacter	g__Colidextribacter	g__Colidextribacter	f__Oscillospiraceae	g__Akkermansia	g__Akkermansia
Adenosylcobalamin/alpha-ribazole phosphatase	**−0.92**	**0.97**	0.55	**−0.87**	0.48	0.67	**0.83**	**−0.82**	0.70	**−0.90**	**−0.63**	**−0.83**	**−0.90**	**−0.89**	**−0.90**
Adenosylcobinamide kinase	**−0.78**	**0.97**	0.55	**−0.83**	0.63	0.66	**0.83**	−0.68	**0.81**	**−0.82**	**−0.77**	**−0.72**	**−0.86**	**−0.85**	**−0.86**
Adenosylcobinamide-GDP ribazoletransferase	**−0.77**	**0.97**	0.60	**−0.87**	**0.65**	**0.70**	**0.85**	**−0.70**	0.70	**−0.82**	**−0.82**	**−0.75**	**−0.86**	**−0.85**	**−0.86**
Adenosylcobinamide-phosphate guanylyltransferase	**−0.78**	**0.97**	0.55	**−0.83**	0.63	0.66	**0.83**	−0.68	**0.81**	**−0.82**	**−0.77**	**−0.72**	**−0.86**	**−0.85**	**−0.86**
Adenosylcobinamide-phosphate synthase	**−0.80**	**0.97**	0.67	**−0.85**	**0.70**	0.68	**0.90**	**−0.73**	0.70	**−0.82**	**−0.78**	**−0.80**	**−0.83**	**−0.83**	**−0.83**
Adenosylcobyric acid synthase (glutamine-hydrolyzing)	**−0.87**	**0.97**	0.63	**−0.77**	**0.70**	0.57	**0.90**	**−0.73**	**0.81**	**−0.82**	−0.65	**−0.78**	**−0.83**	**−0.83**	**−0.83**
Cob(I)yrinic acid a,c-diamide adenosyltransferase	**−0.82**	**0.97**	0.25	−0.62	0.34	0.32	0.61	−0.62	0.59	**−0.90**	−0.28	−0.63	**−0.90**	**−0.89**	**−0.90**
Cobalt-factor II C(20)-methyltransferase	−0.68	**0.97**	**0.77**	**−0.73**	**0.85**	**0.69**	**0.90**	−0.65	0.67	**−0.79**	**−0.88**	**−0.72**	−0.71	**−0.74**	−0.71
Cobalt-precorrin 5A hydrolase	−0.58	0.82	**0.72**	−0.68	**0.82**	**0.72**	**0.78**	−0.55	0.74	**−0.75**	**−0.93**	−0.58	−0.71	**−0.74**	−0.71
Cobalt-precorrin-4 methyltransferase	−0.58	0.82	**0.72**	−0.68	**0.82**	**0.72**	**0.78**	−0.55	0.74	**−0.75**	**−0.93**	−0.58	−0.71	**−0.74**	−0.71
Cobalt-precorrin-5B (C(1))-methyltransferase	**−0.72**	**0.97**	0.68	**−0.82**	**0.79**	**0.72**	**0.92**	−0.68	0.67	**−0.82**	**−0.90**	**−0.75**	**−0.76**	**−0.79**	**−0.76**
Cobalt-precorrin-6A reductase	−0.68	0.82	0.48	**−0.93**	0.21	**0.80**	0.63	**−0.75**	0.34	**−0.79**	**−0.72**	**−0.77**	**−0.83**	**−0.80**	**−0.83**
Cobalt-precorrin-6B (C(15))-methyltransferase (decarboxylating)	0.60	−0.05	−0.17	0.67	−0.12	−0.51	−0.57	0.50	**−0.81**	0.60	0.55	0.45	0.64	0.67	0.64
Cobalt-precorrin-8 methylmutase	−0.58	0.82	**0.72**	−0.68	**0.82**	**0.72**	**0.78**	−0.55	0.74	**−0.75**	**−0.93**	−0.58	−0.71	**−0.74**	−0.71
Cobyrinate a,c-diamide synthase (glutamine-hydrolyzing)	**−0.82**	**0.97**	0.68	−0.67	**0.83**	0.53	**0.91**	−0.68	**0.81**	**−0.79**	−0.67	**−0.73**	**−0.76**	**−0.78**	**−0.76**
Cobaltochelatase	0.52	−0.87	**−0.82**	0.62	**−0.76**	**−0.70**	−0.71	0.62	−0.29	**0.74**	**0.82**	0.67	0.69	0.72	0.69
Hydrogenobyrinic acid a,c-diamide synthase (glutamine-hydrolyzing)	**−0.82**	**0.97**	0.68	−0.67	**0.83**	0.53	**0.91**	−0.68	**0.81**	**−0.79**	−0.67	**−0.73**	**−0.76**	**−0.78**	**−0.76**
Nicotinate-nucleotide--dimethylbenzimidazole phosphoribosyltransferase	**−0.85**	**0.97**	0.60	**−0.75**	**0.76**	0.55	**0.93**	**−0.72**	**0.81**	**−0.82**	−0.68	**−0.77**	**−0.81**	**−0.83**	**−0.81**
Precorrin-2 C(20)-methyltransferase	−0.68	**0.97**	**0.77**	**−0.73**	**0.85**	**0.69**	**0.90**	−0.65	0.67	**−0.79**	**−0.88**	**−0.72**	−0.71	**−0.74**	−0.71
Precorrin-2 dehydrogenase	**−0.73**	0.67	0.67	−0.58	**0.85**	0.48	**0.87**	−0.57	**0.95**	−0.65	−0.68	−0.62	−0.60	−0.62	−0.60
Precorrin-3B C(17)-methyltransferase	−0.58	0.82	**0.72**	−0.68	**0.82**	**0.72**	**0.78**	−0.55	0.74	**−0.75**	**−0.93**	−0.58	−0.71	**−0.74**	−0.71
Precorrin-3B synthase	−0.27	0.36	−0.20	−0.17	**−0.75**	0.06	−0.12	−0.30	−0.02	−0.09	0.42	−0.23	−0.07	−0.01	−0.07
Precorrin-4 C(11)-methyltransferase	−0.58	0.82	**0.72**	−0.68	**0.82**	**0.72**	**0.78**	−0.55	0.74	**−0.75**	**−0.93**	−0.58	−0.71	**−0.74**	−0.71
Precorrin-6A reductase	−0.68	0.82	0.48	**−0.93**	0.21	**0.80**	0.63	**−0.75**	0.34	**−0.79**	**−0.72**	**−0.77**	**−0.83**	**−0.80**	**−0.83**
Precorrin-6B C(5,15)-methyltransferase (decarboxylating)	−0.68	**0.97**	0.67	**−0.83**	**0.69**	**0.74**	**0.80**	−0.65	0.67	**−0.82**	**−0.88**	**−0.70**	**−0.86**	**−0.85**	**−0.86**
Precorrin-8X methylmutase	−0.58	0.82	**0.72**	−0.68	**0.82**	**0.72**	**0.78**	−0.55	0.74	**−0.75**	**−0.93**	−0.58	−0.71	**−0.74**	−0.71
Sirohydrochlorin cobaltochelatase	−0.57	0.67	**0.78**	−0.62	**0.90**	0.61	**0.79**	−0.50	**0.81**	−0.65	**−0.87**	−0.57	−0.64	−0.66	−0.64
Sirohydrochlorin ferrochelatase	**−0.73**	0.67	0.67	−0.58	**0.85**	0.48	**0.87**	−0.57	**0.95**	−0.65	−0.68	−0.62	−0.60	−0.62	−0.60
Threonine-phosphate decarboxylase	**−0.88**	**0.97**	**0.73**	−0.68	**0.70**	0.59	**0.90**	**−0.78**	**0.81**	**−0.82**	−0.62	**−0.82**	−0.71	**−0.73**	−0.71
Uroporphyrinogen-III C-methyltransferase	**−0.72**	**0.97**	0.68	**−0.82**	**0.79**	**0.72**	**0.92**	−0.68	0.67	**−0.82**	**−0.90**	**−0.75**	**−0.76**	**−0.79**	**−0.76**

**Table 5 biomedicines-12-01280-t005:** Spearman correlations evaluated for the representation of bacteria species in the microbiota of *db*/*db* mice and enzymes for cobalamin biosynthesis according to the data from metagenome sequencing. Green font shows positive correlations (*p* < 0.05), red font shows negative correlations (*p* < 0.05), black font shows nonsignificant correlations (*p* > 0.05).

	s__Burkholderia_sp.	s__Muribacter_muris	s__Pseudomonas_sp.	f__Lachnospiraceae; g__uncultured; s__uncultured_bacterium	g__Colidextribacter; s__uncultured_bacterium
Cobalt-precorrin-6A reductase	−0.43	0.65	0.33	**0.84**	0.75
Precorrin-6A reductase	−0.43	0.65	0.33	**0.84**	0.75
Precorrin-2 dehydrogenase	−0.77	**0.88**	0.11	0.23	**0.79**
Uroporphyrinogen-III C-methyltransferase	−0.77	**0.88**	0.11	0.23	**0.79**
Precorrin-2 C(20)-methyltransferase	−0.58	0.74	0.54	**0.78**	**0.79**
Precorrin-3B C(17)-methyltransferase	−0.52	0.74	0.71	**0.84**	0.61
Precorrin-6B C(5,15)-methyltransferase (decarboxylating)	−0.67	0.74	0.71	**0.74**	0.64
Precorrin-4 C(11)-methyltransferase	−0.52	0.74	0.71	**0.84**	0.61
Cobalt-factor II C(20)-methyltransferase	−0.58	0.74	0.54	**0.78**	**0.79**
Precorrin-6A synthase (deacetylating)	−0.21	0.12	**−0.93**	−0.35	0.20
Cobalt-precorrin-5B (C(1))-methyltransferase	−0.67	0.74	0.65	**0.74**	0.61
Cobalt-precorrin-4 methyltransferase	−0.52	0.74	0.71	**0.84**	0.61
Nicotinate-nucleotide--dimethylbenzimidazole phosphoribosyltransferase	**−0.86**	0.77	0.37	0.63	0.50
Cob(I)yrinic acid a,c-diamide adenosyltransferase	−0.73	**0.88**	0.07	0.51	0.64
Adenosylcobinamide kinase	**−0.82**	0.77	0.31	0.56	0.54
Adenosylcobinamide-phosphate guanylyltransferase	**−0.82**	0.77	0.31	0.56	0.54
Adenosylcobinamide-GDP ribazoletransferase	**−0.86**	0.77	0.37	0.63	0.50
Cobalt-precorrin 5A hydrolase	−0.52	0.74	**0.77**	**0.84**	0.64
Threonine-phosphate decarboxylase	−0.67	0.74	0.65	**0.74**	0.61
Sirohydrochlorin cobaltochelatase	−0.52	0.74	0.71	**0.84**	0.61
Sirohydrochlorin ferrochelatase	−0.77	**0.88**	0.17	0.23	**0.82**
Cobalt-precorrin-8 methylmutase	−0.52	0.74	0.71	**0.84**	0.61
Precorrin-8X methylmutase	−0.52	0.74	0.71	**0.84**	0.61
Adenosylcobinamide-phosphate synthase	−0.67	0.74	0.65	**0.74**	0.61
Adenosylcobyric acid synthase (glutamine-hydrolyzing)	−0.67	0.74	0.65	**0.74**	0.61
Cobyrinate a,c-diamide synthase (glutamine-hydrolyzing)	−0.52	0.74	0.71	**0.84**	0.61
Hydrogenobyrinic acid a,c-diamide synthase (glutamine-hydrolyzing)	−0.52	0.74	0.71	**0.84**	0.61
Cobaltochelatase	0.09	0.12	**−0.85**	**−0.20**	−0.07

## Data Availability

The data presented in this study are available upon request from the corresponding author.

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
