# Peer review of "The Obesogenic Gut Microbiota as a Crucial Factor Defining the Depletion of Predicted Enzyme Abundance for Vitamin B12 Synthesis in the Mouse Intestine"

_biomedicines, 2024, doi:10.3390/biomedicines12061280_

Round 1
Reviewer 1 Report
Comments and Suggestions for Authors
It is an interesting paper using both C57BL/6SPF mice and db/db mice studying the gut microbiome difference. However, I have some comments for this manuscript.
1. May I ask the reason of using SPF mice for the gut microbiome studies? Can you please mention the sex of your animals in the method part?
2. In the method part, you mentioned you used a 30 kcal% fat diet as the high-fat diet. May I ask why you choose 30 kcal% fat diet (Altromin C1090-30) as the high-fat diet? It only contains 30 kcal% fat. Typically, a high-fat diet is considered for over 35 kcal% diet for rodents.
3. Can you please let me know why you choose Altromin 1324 as the control diet? I searched your two diets online, and it turns out the two diets have different fiber levels. Moreover, Altromin C1090-30 is a purified diet, while your standard diet (Altromin 1324) is a grain-based chow diet. Grain-based chow diet uses total different ingredients comparing to purified diet, especially the fiber sources and fiber types, which affect the gut microbiome composition. Keeping the same fiber sources and fiber levels is very important for gut microbiome studies.
I would recommend doing another study using high-fat and low-fat diets with the same fiber level and fiber sources.
Reviewer 2 Report
Comments and Suggestions for Authors
This is an interesting study, well-structured, beginning from the baseline knowledge to the most up-graded. Authors used 3 groups of mice: C57BL/6SPF fed with a nornal diet or high-fat diet and db/db mice [leptin receptor deficient], in order to investigate the role of obesity and the disturbed microbiota [due to obesity] on the depleted production of cobalamin [vitamin B12] in the intestine.
Initially, they compare the weight gain between C57BL/6SPF mice fed with a high-fat diet and db/db mice over time, as well as the serum levels of Insulin, Leptin, and Adiponectin vs control. They conclude that db/db mice became more obese than high-hat treated, the finding followed by greater levels of obesity-related hormones and significantly lower alpha-diversity indexes.
After having confirmed the genetically determined character of obesity in db/db mice, they found that there was also a decrease in the representation of the cobalamin synthesis pathways in db/db mice and C57Bl6 high-fat fed vs controls. Namely, the representation of the six pathways for vitamin B12 synthesis in db/db mice and the four pathways in C57Bl6-high fat mice was decreased; the same was true for the enzymes involved in cobalamin symthesis, the finding directly related with the depletion of microbiota, mainly in the db/db mice.
Comments:
1. material are a little confused. they fed mice for 14 or 90d [line 86] but, in the progress of the study it is not clear where the use this sub-groups [were them 5 mice groups?]. I suggest a scheme, like a flow chart, to explain the treatments and the use of animals in each one examination
2. Figure 5 and 6. Better resolution is needed
3. Table 1 and 2. should be aesthetically improved
4. Table 4. Bacteria species names are difficult to read. I suggest the supplement "uncultured bacterium" to be omitted from the headings and explained in the legent only ["all bacteria species were uncultured bacteria, exept of Enterococcus spp and L. plantarum"]
Reviewer 3 Report
Comments and Suggestions for Authors
This manuscript investigates the link between the gut microbiota and vitamin B12 reduction observed in obese patients.
To this aim, 2 different models of obesity (diet-induced and genetically-determined) were used and 16S rRNA sequencing was carried out. Fecal microbiome composition was different between the 3 tested groups and picrust analysis identified significant alterations in the pathways involved in vitamin B12 metabolism. This finding support the hypothesis that vitamin B12 reduction observed in obesity ma be related to the alteration of the obesogenic microbiota.
The topic is interesting and focus the attention on a possible metabolic effect of the well-known gut microbiota alterations related to obesity.
However, some points need to be clarified. First, the association between the obesogenic microbiota and the vitamin B12 metabolism is just predicted by a bioinformatic algorithm and no molecular or functional data are provided to confirm this prediction. This should be clearly stated in the discussion and conclusion sections.
Next, it is not clear what the authors mean as "representation". This word is used several times (also in the title) and is a little bit confounding in my opinion. They mean instead "expression"? Please clarify and choice a more appropriate one.
Figures should be uniformed, it will be better to use in all figures the same colors to identify each group.
Figure 6 is not readable.
Tables are not easy to interpret. In table 4 and 5, no significant correlation may be represented with no color in the cells. A possibility may be to eliminate colors by all cells and use a different color code for the text.
In table 2 is not immediately evident that same pathways are common, so it will be better to put them on the same line.
Also Table 1 should be better structured to compare the results obtained by the 3 different comparisons.
Comments on the Quality of English LanguageMinor english issue need to be fixed.
Reviewer 4 Report
Comments and Suggestions for Authors
The study is interesting, but there are inconsistencies in the description of the animal experiments. For example, there are discrepancies in the total number of animals used and the number of animals in each group, which is crucial for animal experiments. Some places mention 3 groups of animals (n=10), while in others, n=40 is mentioned. It's unclear how the mice are grouped, whether it's BL/SD, BL/SPF, BL/HFD, DB/SD, or db/db.
Moreover, how was it ensured that BL/HFD indeed reached the level of obesity? Testing only insulin, leptin, and adiponectin in serum may not accurately reflect the mice's obesity levels. Why weren't the two most common tests for determining metabolic health, such as the glucose tolerance test (GTT) and insulin tolerance test (ITT), performed?
Additionally, since the focus of the study is on B12 metabolism (production), why wasn't the level of B12 tested?
Furthermore, many of the findings in this study may be attributed solely to the high-fat diet rather than to obesity. It's suggested that the authors compare db/db mice fed a regular diet and a high-fat diet, as this should yield more intriguing data. However, the study lacks an experimental group of db/db mice fed a regular diet.
specific points:
1. In the introduction section, since the current study aims to evaluate the abundance of enzymes and metabolic pathways for vitamin B12 synthesis in the gut microbiota of mouse models of alimentary and genetically determined obesity, I recommend that the authors provide a thorough explanation of previous studies regarding the roles of B12 in obesity.
2. How was it ensured that BL/HFD indeed reached the level of obesity? Generally, compared to the control group, a weight gain of at least 20% is considered indicative of obesity. However, based on your results (in Figure 3a, Changes in mouse body weight during the course of the experiment), the variability in body weight among the SD mice (BL/HFD mice group) suggests that some mice may not have reached obesity. It is suggested that you present the data in terms of weight gain to address this issue.
3. Regarding the previous question, although you have indicated that the mean weight of the mice in the control group was 28.5 ± 5g, and in the high-fat diet-fed mice group was 34.7 ± 1.4g, resulting in a weight difference close to 20%, it is important to clarify that according to past studies on obese mice, the weight difference from the normal group typically exceeds 20%, although there is currently no consistent standard. Please clarify.
4. Please clearly describe the experimental groups as there seems to be ambiguity in the numbers. Some sections mention a total of 40 mice, while in the materials and methods and results sections, it is stated that there are only 3 groups of mice (n=10; totally n=30). As this is the second version of the review, it is expected that the number of animals should be clearly defined. Inconsistent reporting of animal numbers in a study involving animal research is rare and does not meet the basic requirements of animal experimentation.
5. In figure 4 (revised version), it is mentioned that solely testing insulin, leptin, and adiponectin in serum may not accurately reflect the mice's insulin resistance levels. Why didn't you perform the two most common tests for determining metabolic health in mice, namely the glucose tolerance test (GTT) and insulin tolerance test (ITT)? The ITT and GTT tests could be utilized to demonstrate the obese status of mice groups.
6. In figure 4 (revised version), I believe that solely testing insulin, leptin, and adiponectin in serum may not accurately reflect the mice's insulin resistance levels. Why didn't you perform the two most common tests for determining metabolic health in mice, namely the glucose tolerance test (GTT) and insulin tolerance test (ITT)? The ITT and GTT tests could be used to demonstrate the obese status of mice groups.
7. What’s your criteria regarding the successfully established an obese mice model? Please clarify. I don’t think you have successfully established the obese mice model (BL/HFD group). Please clarify and provide the criteria regarding your criteria of obese mice model
8. It's well recognized that the gut microbiota can be significantly influenced by various factors, including dietary control. Therefore, in the findings presented in figures 4 to 6, how can you confirm that the differences observed between BL/SD vs BL/HFD or BL/SD vs db/db (HFD) were truly influenced by the obese status? Firstly, it seems that an obese model was not properly established in the BL/HFD groups. Secondly, it could be argued that most of your findings might be solely attributed to the high-fat diet. Please provide clarification on this matter. While the data in table 3 may partially support the importance of obesity but not HFD on vitamin B12 synthesis pathways, the comparison was made between BL/SD and db/db (HFD). In my opinion, the metabolic background between BL and db/db mice is considerably different, making it potentially unsuitable to compare BL/SD vs db/db (HFD). Why was an animal study not conducted using db/db mice fed a regular diet and a high-fat diet? In other words, comparing the data between db/db (regular diet) vs db/db (HFD) is essential for the current study design. Please clarify this point.
9. It is possible to determine B12 levels in various samples, such as serum, urine, adipose tissue, or tissue samples. Have you considered measuring B12 levels? Having this data could indirectly corroborate your findings on metabolism.
10. Line 47-51: There are two instances of the word "furthermore" here. Please revise to improve the fluency of the descriptions.
11. In this manuscript, both the terms "regular diet" and "standard chow diet" have been used. Although they refer to the same diet, I suggest that the authors use the same term consistently to avoid confusing the readers.
12. In line 78, it is indicated as "db/db (n = 20, males)", however, in other areas, this group of mice is indicated as n=10. Please revise or clarify this discrepancy.
13. In figure 4: The figure legend indicates that "ANOVA of mouse body weight in different experimental groups did not reveal any differences after 4 weeks of feeding". It is unusual to indicate no significance regarding the 4 weeks of feeding. I assume the experiment was completed after 90 days. Please revise or clarify.
Round 2
Reviewer 1 Report
Comments and Suggestions for Authors
Hello,
Thanks for your reply and revision.
It is interesting that your animals did not like to eat higher fat diet and even had steatorrhea. Don't you think it is because they are SPF animals? Previous reports mentioned that SPF animals are more sensitive to environmental changes.
Talking about diets, as you may know, different fiber levels or fiber sources affects the gut microbiome distribution. Since your study tried to compare the gut microbiome composition between control SD-fed group and HFD-fed groups, we need to minimize the mediators - fibers.
Moreover, I checked the Altromin website regarding the two diets. Altromin 1324 mentioned 6.1% crude fiber. The crude fiber level is different from the total fiber level. The crude fiber is based on a very old method of determining fiber, which misses a lot of actual fiber. Typically it accounts for around 4 fold less fiber than the methodology that is used for the total fiber determination. The crude fiber level is different from the total fiber level. The crude fiber is based on a very old method of determining fiber, which misses a lot of actual fiber. Typically it accounts for around 4 fold less fiber than the methodology that is used for the total fiber determination. And Altromin C1090-30 is a purified diet, that said, the crude fiber was not measured. Instead, the crude fiber was calculated, and is the same as total fiber level. That said, your control diet contains much higher fiber level comparing to your 30 kcal% fat purified diet.
Furthermore, for purified diets, it typically only includes cellulose as the only fiber source, which is totally different from grain-based chow diet. In grain-based chow diet, like the control diet you use, it contains a lot of soluble fibers from ingredients, such as soybean meal. Not only for fiber sources, your control diet and 30 kcal% fat diet are in other different ingredients as well. Please note grain-based chow diet is typically used for a healthy control group or for something not related to diet-induced disease models. Unfortunately, the matched diets are very important for studies that need to analyze gut microbiome. The unmatched control diet can confounder your study.
Reviewer 3 Report
Comments and Suggestions for Authors
The authors have accomplished all my previous comments. The manuscript has been improved and I think that this version can be considered for publication.
Author Response
Thank you for the analysis and review of our work.
Reviewer 4 Report
Comments and Suggestions for Authors
The revised version of the manuscript has been considerately improved.
Round 3
Reviewer 1 Report
Comments and Suggestions for Authors
Thanks for sending the information of two diets. Again, the two diets you used in this study are totally different. Altromin 1324, as their website states, is a grain-based chow diet, which uses soy, wheat, corn, etc. Moreover, each ingredient, such as soymeal, can contribute to fat, protein, carbohydrate, and fiber. However, your high-fat diet is not a grain-based chow diet, which uses refined ingredients. The crude fat analysis for grain-based chow diet was not accurate at all because crude fiber is the insoluble portion of the cell wall of plants, while grain-based chow diets contain a lot of soluble fiber. Instead, purified diets, such as your high-fat diet, mainly only contain dietary fiber from cellulose. Thus, although the website said Altromin 1324 contains 6.1% crude fiber, the real dietary fiber level is much higher than this. Again, different fiber level and fiber sources can shift gut microbiome composition easily.
I'm totally with you if you want to consider the grain-based chow diet as a healthy reference group for your study. However, your gut microbiome data should not be included as it is not reliable.